# Distinct Roles of Ena ATP Family Proteins in Sodium Accumulation, Invasive Growth, and Full Virulence in *Colletotrichum gloeosporioides*

**DOI:** 10.3390/jof9050566

**Published:** 2023-05-13

**Authors:** Tian-Ci Deng, Ji-Yun Yang, Mei-Ling Sun, Yun-Zhao Zhang, Yun-Ting Pan, Lin Huang

**Affiliations:** 1Co-Innovation Center for Sustainable Forestry in Southern China, Nanjing Forestry University, Nanjing 210037, China; dtc919421@163.com (T.-C.D.); yangjiyun2018@163.com (J.-Y.Y.); sunmeiling0426@163.com (M.-L.S.); 15751455333@163.com (Y.-Z.Z.); panyuting1997@gmail.com (Y.-T.P.); 2College of Forestry, Nanjing Forestry University, Nanjing 210037, China

**Keywords:** *Colletotrichum gloeosporioides*, Ena ATPase proteins, subcellular localization, sodium accumulation, pathogenicity

## Abstract

*Colletotrichum gloeosporioides*, a significant fungal pathogen of crops and trees, causes large economic losses worldwide. However, its pathogenic mechanism remains totally unclear. In this study, four Ena ATPases (*Exitus natru*-type adenosine triphosphatases), homology of yeast Ena proteins, were identified in *C. gloeosporioides*. Gene deletion mutants of Δ*Cgena1*, Δ*Cgena2*, Δ*Cgena3*, and Δ*Cgena4* were obtained through the method of gene replacement. First, a subcellular localization pattern indicated that CgEna1 and CgEna4 were localized in the plasma membrane, while the CgEna2 and CgEna3 were distributed in the endoparasitic reticulum. Next, it was found that CgEna1 and CgEna4 were required for sodium accumulation in *C. gloeosporioides*. CgEna3 was required for extracellular ion stress of sodium and potassium. CgEna1 and CgEna3 were involved in conidial germination, appressorium formation, invasive hyphal development, and full virulence. The mutant of Δ*Cgena4* was more sensitive to the conditions of high concentrations of ion and the alkaline. Together, these results indicated that CgEna ATPase proteins have distinct roles in sodium accumulation, stress resistance, and full virulence in *C. gloeosporioides*.

## 1. Introduction

In living cells, sodium and potassium are strictly controlled within certain concentration range [1]. Maintaining appropriate ion concentration requires the participation of P-type ATPases [2,3]. P-type ATPases are a large family of membrane proteins that play important roles in ion transport across the plasma membrane and form a typical phosphatase intermediate which hydrolyzes ATP to resist the electrochemical gradient [4,5]. Two P-types of ATPases, H^+^-ATPase and Ena ATPase, were reported in fungi, which are involved in maintaining the ion balance inside and outside the cell [3]. H^+^-ATPase, an ATP-driven proton pump, forms an electrochemical gradient on the plasma membrane which coordinates the absorption of K^+^ and the extrusion of Na^+^ through the secondary transport systems, resulting in the asymmetric distribution of K^+^ and Na^+^ [6,7]. Ena ATPase can combine ATP hydrolysis with cation (K^+^, Na^+^ and Li^+^) transports to regulate the electrochemical potential gradient [8,9]. Although both P-type ATPases are complementary, the deletion of ATPase genes suggests that the Ena ATPases are the major determinant for sodium tolerance under standard growth conditions in *Saccharomyces cerevisiae* [10].

An Ena ATPase was first identified in *S. cerevisiae* [11,12]. The efflux of sodium, lithium and potassium was found [10,13]. There are different copies of *ENA* genes which usually encode the same or similar proteins in fungi as well as bryophytes and protozoa [3]. In *S. cerevisiae*, five Ena ATPase genes (*ENA1*, *ENA2*, *ENA3*, *ENA4*, and *ENA5*) were identified [14]. Different Ena ATPases have specificity for the ion transport of K^+^ and Na^+^ [12]. When *S. cerevisiae* were exposed to high Na^+^ stress, the expression of ScEna1 was dramatically increased. The deletion of gene *ScENA1* affects the efflux of Na^+^. Deletion of *ScENA2* led to more sensitive to high pH, and the decreased efflux of K^+^, however, it did not alter the tolerance to Na^+^ [12,15]. Ena21 and Ena22 of the Ena family were identified in *Candida albicans* and *Candida dubliniensis*, respectively. Both were identified as being involved in the regulation of salt tolerance [16]. CnEna1 played a role in virulence of *Cryptococcus neoformans* [17]. In *Aspergillus fumigatus*, the expression of AfEnaA was rapidly up-regulated after exposure to the alkaline condition. Additionally, the mycelial growth of the gene deletion mutant was significantly depressed under the high concentration of sodium, divalent manganese ions and alkaline conditions [18]. Although Ena proteins in yeast and filamentous fungi were extensively studied, the function of Ena proteins in plant pathogenic fungi remain largely unknown.

*Colletotrichum gloeosporioides* is a semi-biotrophic fungus that infects a variety of plant hosts [19]. The *C. gloeosporioides* genome data were released, and provides a reference for functional gene identification and control target discovery [20]. In *C. gloeosporioides*, potassium transporter-related protein CgSat4 was identified in previous studies. CgSat4 was involved in K^+^ uptake by regulating the localization and expression of the potassium transporter CgTrk1, and is required in cell wall integrity, hyperoxide stress response, and pathogenicity [21]. The Δ*Cgppz1* mutant was hypersensitive to osmotic stresses, cell wall stressors and oxidative stress [22]. However, despite these advances, it remains unknown whether the Ena ATPase proteins participate in the accumulation of Na^+^ and K^+^, and plant infection in *C. gloeosporioides*.

This study investigated the subcellular localization of four Ena ATPase proteins CgEna1, CgEna2, CgEna3, and CgEna4 in *C. gloeosporioides.* CgEna1 and CgEna4 were required for the accumulation of sodium, rather than potassium. CgEna3 was required for the extracellular ion stress resistance of sodium and potassium. CgEna1 and CgEna3 were involved in conidial germination, appressorium development, invasive hyphal growth, and full virulence. CgEna4 was required for cation stress resistance under non-acidic conditions. These findings can reveal the potential mechanisms of ion transport and the roles of Ena proteins in the phytopathogenic fungi.

## 2. Results

### 2.1. Ena Family Proteins in C. gloeosporioides and Their Localization Pattern

Four Ena family protein homologues in *C. gloeosporioies* were identified with the *S. cerevisiae* Ena family proteins as the reference to search the *C. gloeosporioides* genome database, and were named CgEna1, CgEna2, CgEna3, and CgEna4. CgEna1, CgEna2, CgEna3, and CgEna4 shared a high conserved sequence and contained a homologous ATPase catalytic domain (Appendix A).

To examine the localization pattern of CgEna1, CgEna2, CgEna3, and CgEna4, the *GFP* gene was fused to the C-terminal of each gene. Each construct of the four genes was transformed into the corresponding gene deletion mutant, which was used for localization pattern analysis of the target gene. Observation with a microscope showed that the GFP signal of CgEna1-GFP and CgEna4-GFP was strongly deposited on the plasma membrane of *C. gloeosporioides* (Figure 1A). For excluding the interference of the cell wall, a similar localization pattern was observed in the plasma membrane in their protoplasts. These results indicated that both CgEna1 and CgEna4 were expressed in the plasma membrane of *C. gloeosporioides*. However, the GFP signal of CgEna2-GFP was mainly localized in the endoplasmic reticulum and CgEna3-GFP completely in the endoplasmic reticulum. To further confirm subcellular localization, the fusion plasmid of *CgLHS1*-*RFP* was transferred into *C. gloeosporioides* in which CgEna2-GFP or CgEna3-GFP was expressed. Lhs1 has been employed as a marker protein of the endoplasmic reticulum in phytopathogenic fungi [23]. The co-expression of CgLhs1-RFP was merged with CgEna2-GFP or CgEna3-GFP, which were distributed in an endoplasmic reticulum localization pattern (Figure 1B). These results indicated that CgEna1, CgEna2, CgEna3, and CgEna4 had different localizations.

### 2.2. Ena Family Proteins Expressed at the Early Stage of Plant Infection

The transcription analysis showed that, besides *CgENA1*, the expression level of *CgENA2*, *CgENA3* and *CgENA4* in conidia was significantly higher than mycelia (Appendix A). The expression levels of *CgENA1*, *CgENA2*, *CgENA3* and *CgENA4* was significantly induced during the early interaction of *C. gloeosporioides* and its host plant. At the initiated infected phase of 4 to 8 hpi, *CgENA1* exhibited 5- to 8-fold greater expression than the mycelium stage. Furthermore, transcripts of *CgENA2*, *CgENA3*, and *CgENA4* were significantly increased in most invasion periods (Appendix A). To investigate the function of CgEna1, CgEna2, CgEna3, and CgEna4, we obtained the gene deletion mutants of Δ*Cgena1*, Δ*Cgena2*, Δ*Cgena3*, Δ*Cgena4* by replacing their coding region with the hygromycin resistance cassette (*HPH*) (Appendix A). These deletion mutants have been confirmed by PCR (Appendix A). These corresponding complemented strains were generated by reintroducing the target gene ORF and its native promoter into the corresponding gene deletion mutants. Vegetative growth assays showed that no significant difference was observed among the mutants of Δ*Cgena1*, Δ*Cgena2*, Δ*Cgena3*, or Δ*Cgena4*, compared to wild type SMCG1#C and the corresponding complemented strains (Appendix A). These results suggested that CgEna1, CgEna2, CgEna3 and CgEna4 play crucial roles at the early stage of plant infection, while they were not indispensible for vegetative growth in *C. gloeosporioides*.

### 2.3. CgEna1 and CgEna4 Are Involved in Sodium Accumulation

In the yeast *S. cerevisiae*, Ena proteins are cationic pumps that transport sodium to the outside of cells and transport potassium into cells [8]. In the study, the concentration of sodium and potassium in the mycelia of the wild type and mutants of Δ*Cgena1*, Δ*Cgena2*, Δ*Cgena3*, and Δ*Cgena4*, were measured. This result showed that deletion of *CgENA1*, *CgENA2*, *CgENA3* and *CgENA4* had no significant effects on the content of mycelial potassium (Figure 2A). The sodium concentration of the mutants of Δ*Cgena2* and Δ*Cgena3* were similar to that of the wild type. However, there was significantly higher sodium content in the mycelial in the Δ*Cgena1* and Δ*Cgena4*, compared to the wild type (Figure 2B). In addition, the extracellular ion stress response of the strains was examined. Interestingly, only the growth of the Δ*Cgena3* mutant was significantly inhibited when exposed to extracellular Na^+^ or K^+^ (Figure 3A,B). These results suggest that CgEna1 and CgEna4 are involved in regulating sodium accumulation in *C. gloeosporioides*.

### 2.4. CgEna1 and CgEna3 Indispensable for Full Virulence

The pathogenicity of the wild type, Δ*Cgena1*, Δ*Cgena2*, Δ*Cgena3*, Δ*Cgena4*, and corresponding complemented strains were measured on healthy leaves of *C. lanceolata* and *L. chinense* × *tulipifera*, respectively. At 5 or 7 days post inoculation(dpi), there was no significant difference among the diameter of lesions caused by Δ*Cgena2*, Δ*Cgena4*, and the wild type. However, the lesions caused by Δ*Cgena1* and Δ*Cgena3* mutants were significantly smaller than those of other tested strains (Figure 4A,C). Similarly, the Δ*Cgena1* and Δ*Cgena3* mutants were also less virulent on *L. chinense* × *tulipifera* (Figure 4B,D). These results suggest that CgEna1 and CgEna3 are required for the full virulence of *C. gloeosporioides*.

### 2.5. CgEna1 and CgEna3 Regulate Conidial Germination, Appressorium Formation and Invasive Hypha Development

Conidial germination of the wild type, Δ*Cgena1*, Δ*Cgena2*, Δ*Cgena3*, Δ*Cgena4*, and corresponding complemented strains were measured at 2 and 4 hpi. These data showed that there was no significant difference among the wild type, Δ*Cgena2*, and Δ*Cgena4*. However, compared with the wild type, the conidial germination ratio of Δ*Cgena1* was significantly decreased at 2 hpi (Figure 5A). In the mutant of Δ*Cgena3*, the conidial germination ratio was significantly lower than that of the wild type (Figure 5A). These data indicate that the deletion of the *CgENA1* and *CgENA3* significantly delayed conidial germination at the early stage. Similarly, the appressorium formation ratio of the mutants of Δ*Cgena1* and Δ*Cgena3* was also decreased significantly, while no changes happened to Δ*Cgena2* and Δ*Cgena4* mutants (Figure 5B).

The invasive hyphae (IH), developed on the onion epidermis, was further observed 24 h after inoculation. The data showed that the most IH of the wild type belonged to type III and IV. Compared to wild type strain SMCG1#C, the mutants of Δ*Cgena1* and Δ*Cgena3* produced more IH of type I and II and less IH of type III and IV proportionally (Figure 5C). Collectively, all results indicated that CgEna1 and CgEna3 are indispensable for full virulence by involving in the development of the invasive structure in *C. gloeosporioides*.

### 2.6. CgEna4 Is Required for Ion Stress Response under Certain Extracellular Conditions

It has been reported that the expression of *ENA* genes was induced by alkaline conditions in the yeast [10]. In this study, it was found that the deletion of Cg*ENA1*, Cg*ENA2*, Cg*ENA3*, andCg*ENA4*, respectively, had no significant effects on the vegetative growth on the CM (complete medium) plates. Under the sole stress, like NaCl, KCl, or non-acidic conditions, the growth of Δ*Cgena4* wasn’t inhibited (Figure 6A,B,D,E). However, when exposed to 0.7 M NaCl under the pH 7 or 10 conditions, the vegetative growth of the Δ*Cgena4* mutant was significantly depressed compared to the wild type (Figure 6C,F). Interestingly, under the condition of same pH, the growth inhibition in the Δ*Cgena4* mutant wasn’t observed when the NaCl was replaced by KCl (Figure 6C,F). These results indicated that CgEna*4* plays an important role in the growth of *C. gloeosporioides* under the non-acidic condition combined with sodium ion stress.

## 3. Materials and Methods

### 3.1. Fungal Strains and Culture Conditions

The *C. gloeosporioides* strain SMCG1#C [20], and mutant strains of Δ*Cgena1*, Δ*Cgena2*, Δ*Cgena3*, Δ*Cgena4*, and the corresponding complemented strains were cultured on CM medium at 25 °C [24]. The fungal mycelia were cultured in a liquid CM medium for DNA extraction, protein extraction and protoplasts preparation, as described previously [22].

### 3.2. Gene Deletion of CgENA1, CgENA2, CgENA3, and CgENA4

The complete gene sequences of Ena ATPases, Cg*ENA1*, Cg*ENA2*, Cg*ENA3* and Cg*ENA4* of *C. gloeosporioides* were obtained from the genome database of GenBank (https://www.ncbi.nlm.nih.gov/). The upstream and downstream fragments of each gene were amplified from the wild type gDNA, and ligated to two ends of the hygromycin phosphotransferase (*HPH*) gene, respectively, using the overlapping PCR method [21]. The fusion fragments were amplified and transformed into the protoplasts of the wild type as described previously [25,26]. The putative gene deletion transformants were confirmed by PCR assays.

### 3.3. Complementation of Targeted Gene Mutants

To obtain the complemented strains of the genes mutants, the fragments containing the entire ORF (open reading frame) of each targeted gene and its native promoter were amplified and inserted into a pYF11 vector and fused to the GFP reporter gene [27]. The resulting construct of each gene was verified by sequencing and reformatting it into the corresponding gene mutant. The candidate complementation transformants were verified by detecting the GFP signals, as described previously [28].

### 3.4. RNA Isolation and Quantitative Real-Time PCR (qRT-PCR)

To determine the relative abundance of *CgENA1*, *CgENA2*, *CgENA3*, and *CgENA4* at the early infection stage, the onion epidermis, inoculated with conidia of the WT, was harvested at 2, 4, 6, 8, 12, 16, 24 and 32 hpi for RNA extraction. Total RNA was extracted through a RNA Extraction kit according to the manufacturer’s instructions (Conway Reagent, Nanjing, China). The first strand of cDNA was synthesized through an RNA reverse transcription reagent (Novozan Biotechnology Co., Nanjing, China). RT-PCRs were carried out according to the method described by [29], and the house-keeping gene *ACTIN* was used as the internal control. The relative expression level of each gene was calculated through the 2^−ΔΔCT^ method as described in [30]. The experiment was carried out three times in each treatment. All used primers are listed in Appendix A.

### 3.5. Assays for Vegetative Growth

Mycelial plugs (2 mm × 2 mm in size) of the wild type, mutants of Δ*Cgena1*, Δ*Cgena2*, Δ*Cgena3*, Δ*Cgena4*, and the corresponding complemented strains were inoculated on PDA (Potato dextrose medium), OMA (Oatmeal agar), MM (Minimal medium) and Mathur’s medium plates, respectively, and cultured at 25 °C in the dark for 5 days. The diameter of fungal colonies was measured as described previously [24]. Three independent biological experiments were performed.

### 3.6. Assays of Stress Resistance and Sodium/Potassium Accumulation

For mycelial sodium and potassium content assays, the wild type, mutants of Δ*Cgena1*, Δ*Cgena2*, Δ*Cgena3*, Δ*Cgena4*, and their corresponding complemented strains, were cultured in a liquid CM medium for 2 days as described previously [21,29]. The mycelial pellets were collected through a miracloth (EMD Millipore, Bellerica, MA, USA) to remove the filtrate. Fungal mycelia were harvested and dried in a freeze dryer. The dried mycelia were digested with sulfuric acid. The potassium and sodium content was then detected by a flame spectrophotometer, respectively, as described previously [21]. Three independent biological experiments were performed.

For stress resistance analysis, the wild type, mutants of Δ*Cgena1*, Δ*Cgena2*, Δ*Cgena3*, Δ*Cgena4*, and their corresponding complemented strains were inoculated on CM plates supplemented with 0.7 M NaCl and 0.7 M KCl, respectively, and cultured at 25 °C in the dark. At 5 dpi, the diameter of fungal colonies was measured [6]. Three independent biological experiments were performed.

### 3.7. Pathogenicity Tests

For pathogenicity tests, the conidium of the wild type, mutants of Δ*Cgena1*, Δ*Cgena2*, Δ*Cgena3*, Δ*Cgena4*, and their corresponding complemented strains were collected, and the concentration of the conidial suspensions of each strain was adjusted to 1 × 10^5^ spores/ml [25]. A total of ten µL of conidial suspensions were inoculated on the healthy leaves of *Liriodendron chinense* × *tulipifera*. An amount of five µl of the conidial suspension of each strain was inoculated on the healthy leaves of *C. lanceolata.* The inoculated leaves were kept in a moist chamber at 25 °C. The size of the lesions was measured and photographed at 7 dpi or 5 dpi, respectively, as described previously [25].

### 3.8. Observation of Conidial Germination, Appressorium Formation and Invasive Hyphal Development

Conidium of the wild type, mutants of Δ*Cgena1*, Δ*Cgena2*, Δ*Cgena3*, Δ*Cgena4* and their corresponding complemented strains were prepared by the method described above. The concentration of conidium was adjusted to 2 × 10^4^ conidia/mL. A total of ten µl of conidial suspensions of each strain were dropped on hydrophobic glass slides (Fisher brand) and placed in an incubator at 25 °C [24]. Conidial germination and appressorium formation was observed under a compound microscope (Carl Zeiss, Jena, Germany) at 2, 4, 8, and 12 h hpi. The experiment was conducted three times, with at least 120 conidium counted per strain.

Invasive hyphae (IH) were induced on the onion epidermal strip according to the method as described previously [25]. IH were divided into four different types: type I, no IH development, type II, IH with one branch, type III, IH with at least two branches, but limited expansion and type IV, IH with numerous branching and extensive hyphal growth. The experiment was performed three times, and at least 50 invasive structures were observed in each treatment.

### 3.9. Statistical Analysis

The mean standard deviation (SD) was calculated for each treatment. The statistical analysis of significant among difference among treatments in this study was performed with a one-way analysis of variance (ANOVA) followed by least significant differences multiple range tests. Statistical analyses were carried out by the SPSS 19.0 software program (SPSS Inc., Chicago, IL, USA).

## 4. Discussion

In many fungal genomes, there are two or more genetically divergent *ENA* genes [3,30]. In this study, four Ena ATPase homologs: CgEna1, CgEna2, CgEna3, and CgEna4 were identified in *C. gloeosporioides* by blastp searches through *S. cerevisiae* Ena ATPase sequences against *C. gloeosporioides*. These four Ena proteins shared high sequence identity with corresponding proteins found in other fungi, particularly within the ATPase domain. It was found that these Ena proteins play critical roles at the early stage of plant infection, while they are not involved in vegetative growth. Among them, CgEna2 was located in the endoplasmic reticulum and was not involved in the regulation of pathogenicity. CgEna1 and CgEna3 were required for the full virulence of *C. gloeosporioides*. CgEna3 was also involved in the response to extracellular Na^+^ and K^+^ stress. CgEna1 and CgEna4 regulated the sodium accumulation. Interestingly, CgEna4 played a critical role in the growth of *C. gloeosporioides* under Na^+^ stress and non-acidic conditions.

The main function of Ena ATPases in fungi has been proven to pump cation [31]. ScEna1, a dual functional protein, is responsible for the extracellular transport of sodium and also accumulation transport of potassium into the cell [15]. Studies of the roles of Ena2 in *Debaryomyces occidentals* showed that the mutation of Ena2 led to the reduction of intracellular potassium accumulation. In *Neurospora crassa*, Ena1 is only involves in efflux Na^+^, while others Ena ATPases regulate the efflux of Na^+^ and K^+^ [3]. In contrast, the deletion of *ENA* genes was found to have no significant effect on potassium among the wild type and all mutants, and higher intracellular sodium content was tested in the mutants of Δ*Cgena1* and Δ*Cgena4*, indicating that CgEna1 and CgEna4 might be involved in the regulation of sodium in *C. gloeosporioides*. This finding further enhances the understanding of the role of Ena proteins in sodium efflux.

The different localization pattern of Ena proteins may determine the functional differentiation [15,30]. In *Aspergillus nidulans*, EnaB located in the endoplasmic reticulum that greatly improved its tolerance to highly toxic lithium [32]. In this study, CgEna2 and CgEna3 were found to be both located in the endoplasmic reticulum, but only Δ*Cgena3* showed hypersensitivity to Na^+^ and K^+^ stresses, indicating that CgEna3 was involved in extracellular ion stress response. A similar function of EnaB was observed in *A. nidulans* [32]. In *S. cerevisiae*, Ena1 was located in the plasma membrane and seems to be a sodium pump. [30]. In agreement with that, CgEna1 and CgEna4 was observed to locate on the plasma membrane in hypha and protoplasts of *C. gloeosporioide*. Δ*Cgena1* and Δ*Cgena4* significantly increased the concentration of Na^+^ in *C. gloeosporioides.* CgEna1 and CgEna4 may regulate the accumulation of sodium.

In the interaction between fungi and their hosts, the fungi have the ability to grow in a complex pH environment. In most cases, Ena ATPases are not required in acidic environments, as neutral Na^+^/H^+^ and K^+^/H^+^ antiporters can replace Ena ATPase [14,32,33]. However, when the extracellular pH increased, the *ScENA* genes were strongly in up-regulation, which causes the loss of the function of the anti-transporter ScNha1, resulting in the inability to correctly excrete sodium and potassium [34,35]. In general, the excessive increase of the intracellular concentration of sodium is harmful for an organism [36]. Under usual growth conditions, the Ena proteins are the main determinant of sodium detoxification tolerance to alkaline ambient pH in *A. nidulans*, [32]. Large amounts of sodium accumulation in the cytosol tend to destroy essential and sensitive enzymes [37]. In this study, the lack of *CgENA4* may affect the maintenance of the redox homeostasis of the transport of sodium in *C. gloeosporioides*. The deletion of *CgENA4* significantly inhibited the mycelial growth under combined conditions of 0.7 M NaCl and pH 7, or pH 10, indicating that non-acidic pH may alter the ion resistance mediated by CgEna4 in *C. gloeosporioides*.

In addition, four Ena proteins were investigated for their potential roles in the pathogenicity of *C. gloeosporioides*. Inoculation assays manifested on mutants of Δ*Cgena1* and Δ*Cgena3*, relative to the wild-type SMCG1#C, developed smaller disease lesion on the leaves of *C. lanceolata* and *L. chinense* × *tulipifera* leaves. The invasive structures of appressorium and invasive hyphae (IH) were also found to be delayed in mutants of Δ*Cgena1* and Δ*Cgena3*. Therefore, it is suggested that CgEna1 and CgEna3 are required for the full virulence of the regulated invasive structure that was developed.

In summary, four Ena proteins were identified in *C. gloeosporioides*, and investigated their potential roles in ion transport, appressorium formation, invasive hyphae development and pathogenicity. These findings help reveal the potential mechanisms of ion transport and the roles of Ena proteins in plant infection in *C. gloeosporioides*.

## Figures and Tables

**Figure 1 jof-09-00566-f001:**
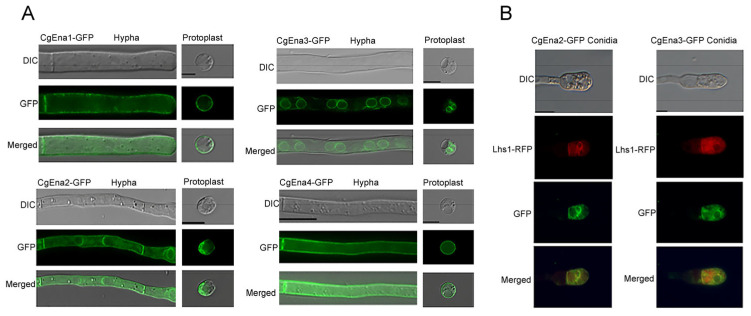
**Localization pattern of Ena family proteins *C. gloeosporioides*.** (**A**) Subcellular localization of CgEna1, CgEna2, CgEna3, CgEna4 in the hypha and protoplast of *C. gloeosporioides*. The fused constructs of *CgEna1-GFP*, *CgEna2-GFP*, *CgEna3-GFP*, *CgEna4-GFP* were transformed into the corresponding gene deletion mutants. The subcellular localization of CgEna1, CgEna2, CgEna3, and CgEna4 was observed in the hypha and protoplast by using a microscope (Zeiss, Oberkochen, Germany). (**B**) Proteins of CgEna2-GFP, CgEna3-GFP, and the endoplasmic reticulum maker protein of Lhs1-RFP were co-expressed in the conidia of *C. gloeosporioides*. Scale bars = 10 μm.

**Figure 2 jof-09-00566-f002:**
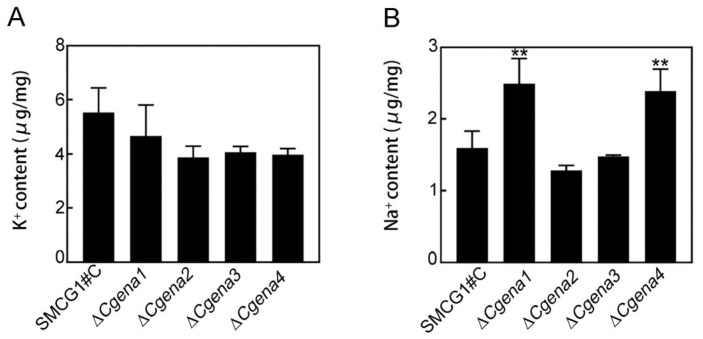
**CgEna1 and CgEna4 are involved in sodium accumulation.** The concentration of potassium (**A**) and sodium (**B**) in the mycelia of the wild type, Δ*Cgena1*, Δ*Cgena2*, Δ*Cgena3*, and Δ*Cgena4*. Error bars represent the SD (standard deviation) and “**” indicate a significant difference at *p* < 0.01. Three independent experiments were performed and similar results were obtained.

**Figure 3 jof-09-00566-f003:**
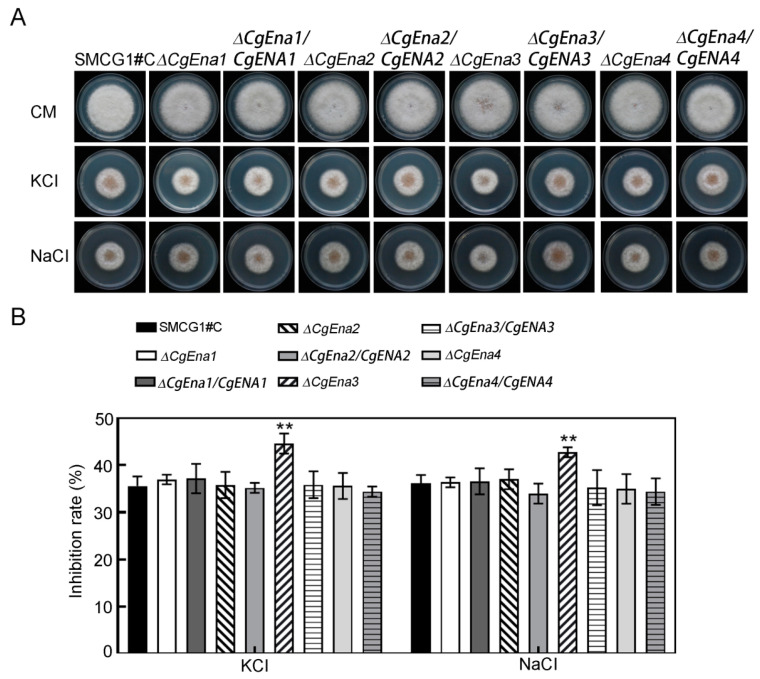
**CgEna3 is involved in ion stress resistance.** (**A**) Colonies of the wild type, Δ*Cgena1*, Δ*Cgena2*, Δ*Cgena3*, Δ*Cgena4*, and their corresponding complemented strains, cultured on CM medium supplemented with 0.7 M NaCl and KCl. (**B**) The growth inhibition rate of the tested strains exposed to stress conditions indicated in (**A**). Error bars represent the SD. “**” indicate significant differences at *p* < 0.01. Three independent experiments were performed, and similar results obtained.

**Figure 4 jof-09-00566-f004:**
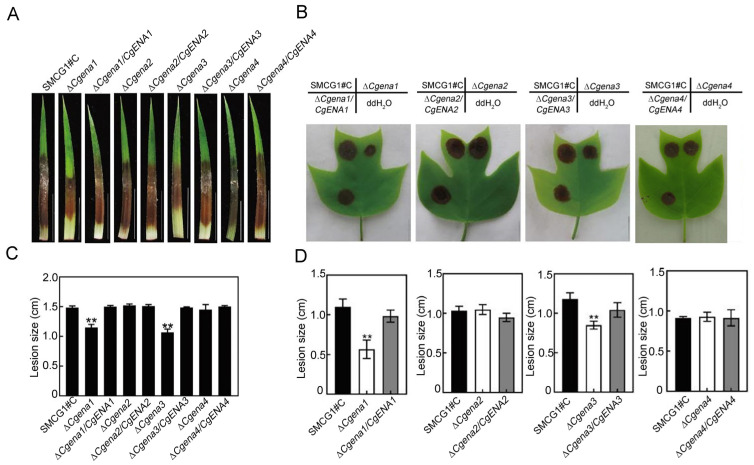
CgEna1 and CgEna3 are required for full virulence on host plants. Lesions developed on leaves of *C. lanceolata* (**A**) and *L. chinensis* × *tulipifera* (**B**) after inoculation by the wild type, Δ*Cgena1*, Δ*Cgena2*, Δ*Cgena3*, Δ*Cgena4*, and their corresponding complemented strains. Pictures were photographed at 5 dpi or 7 dpi. ddH_2_O was used as the negative control. (**D**) Average diameter of lesions developed on leaves of *C. lanceolata* (**C**) and *L. chinensis* × *tulipifera*. Error bars represent the SD. “**” indicate significant differences at *p* < 0.01. Three independent biological experiments were performed and yielded similar results.

**Figure 5 jof-09-00566-f005:**
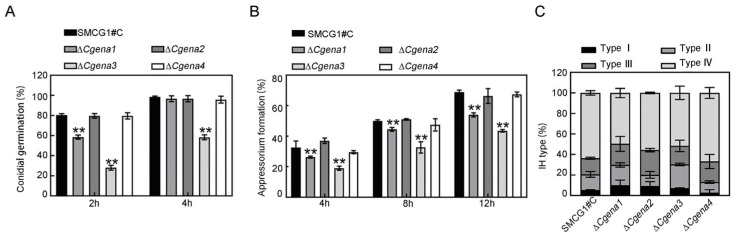
**CgEna1 and CgEna3 regulate conidial germination, appressorium formation, and invasive hyphal development.** (**A**) Conidial germination ratio of the wild type, Δ*Cgena1*, Δ*Cgena2*, Δ*Cgena3* and Δ*Cgena4* at 2 and 4 hpi. (**B**) Appressoria formation ratio of the wild type, Δ*Cgena1*, Δ*Cgena2*, Δ*Cgena3* and Δ*Cgena4* at 4, 8 and 12 hpi. (**C**) Proportion of different types of invasive hypha (IH) developed by the wild type, Δ*Cgena1*, Δ*Cgena2*, Δ*Cgena3* and Δ*Cgena4*. Error bars represent the SD. “**” indicate significant difference at *p* < 0.01. Three independent biological experiments were performed and yielded similar results.

**Figure 6 jof-09-00566-f006:**
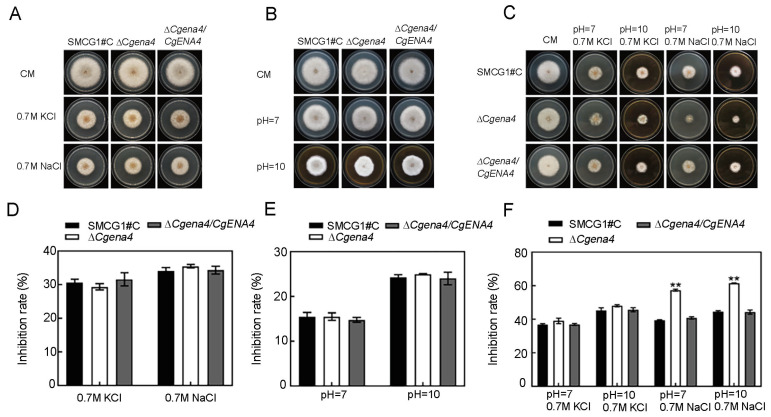
**Roles of CgEna4 on sodium stress response under non-acidic condition.** (**A**) Colonies of the wild type, Δ*Cgena4*, and complementary strain Δ*Cgena4*/*CgENA4* inoculated on CM medium supplemented with 0.7 M NaCl or 0.7 M KCl under normal pH. (**B**) Colonies of the wild type, Δ*Cgena4*, and complementary strain Δ*Cgena4*/*CgENA4* inoculated on the CM medium under different pH (7 or 10). (**C**) Colonies of the wild type, Δ*Cgena4*, and complementary strain Δ*Cgena4/CgENA4* inoculated on CM medium under different combination conditions of pH and sodium stress. (**D**–**F**) The inhibition ratio of vegetative growth of the above tested strains indicated in (**A**–**C**), respectively. Error bars represent the SD. “**” indicate significant difference at *p* < 0.01. Three independent biological experiments were performed.

## Data Availability

The data presented in this study are available on request from the corresponding author.

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
