# Peer review of "Distinct Roles of Ena ATP Family Proteins in Sodium Accumulation, Invasive Growth, and Full Virulence in Colletotrichum gloeosporioides"

_jof, 2023, doi:10.3390/jof9050566_

Round 1

Reviewer 1 Report

The present study addresses the identification, molecular and functional characterization of four ATPases from Colletrotrichum gloeosporioides an important pathogen of agronomic interest. By analysis of gene expression, subcellular localization, knockdown, and molecular complementation were described the functions of ATPases in the formation of conidial germination, appressorium and invasive hyphal growth. The main findings of the manuscript described specific roles of ATPases in C. gloeosporioides ion transport, appressorium development and pathogenicity. Overall, the manuscript is well structured and written. I have only some comments described below:

Structurally the four ATPases showed differ slightly. The main differences with respect to function are justified regarding to subcellular location. As shown in Fig. S2, at the different time points during gene expression analysis “two waves of expression of Ena genes” can be observed. The first one between 4-6 hpi and second one later after 12 hpi. What explain this transcriptional response? A mention should be included about these findings.

Regarding to CgEna in potassium and sodium uptake (Figure 2): I think potassium and sodium mycelia concentration corresponding to complemented Cg strains should be included to improve the results about role of Ena in ions uptake.

In figure 4: It seems to me that the lesions on C. lanceolata and L. chinensis × tulipifera, when infected by ΔCgEna4 strain are more severe than lesions of complemented strains ΔCgEna4/CgEna4 although the lesion size does not differ significantly when compared with WT strain. Please check it.

Pg. 5, Line 178-179. “These data indicated that the deletion of CgEna3 delayed the conidial germination” According to the Fig. 5A, the deletion of CgEna1 gene delayed the conidial germination.

Line 349-351. "In this study, the deletion of CgEna4 significantly inhibited the mycelium growth under combined conditions of 0.7 M sodium and pH 7 or pH 10, in dicating that non-acidic pH may alter the ion resistance mediated by CgEna4 in C.gloeosporioides".

In this sentence the authors could point out in a few words that the lack of ΔCgEna4 function would affect the maintenance of redox homeostasis of transport of sodium in C. gloeosporioides as described in Results (Fig 2).

Minors

“Abstract”

 Line 13, “Ena ATPases” The abbreviations should be spelling out the first time is cited.

“Material and Methods”

Line 238, What mean “CM plates supplemented with 0.7 M NaCl and 0.7 M NaCl, respectively”?

Line 264, “Fungal mycelia were harvested and dried in a freeze dryer” Please describe the procedure to harvest the fungal mycelia.

Author Response

Reviewer #1

We thank you for your helpful comments and suggestions for improving the manuscript. I appreciate the reviewer’s suggestions. The answers to question are as follow.

1 Question: Structurally the four ATPases showed differ slightly. The main differences with respect to function are justified regarding to subcellular location. As shown in Fig.S2, at the different time points during gene expression analysis “two waves of expression of Ena genes can be observed. The first one between 4-6 hpi and second one later after 12 hpi. What explain this transcriptional response? A mention should be included about these findings.

Response: Thank you for your valuable suggestions! Following the comment, the gene expression data of Ena genes showed two waves at 4-6 hpi and 24-36 hpi at Fig.S2. According to our database, transcriptional level of ENA genes were obviously increased at the time of early infection compared to coniduim or hypha. In the time of later infected, ENA genes also increased to a new expressional peak. In our views, it is cause that the condium of C.gloeosporioides were initiated contact with host, and then geminated tube hypha and appressorium to penetrate into plant cell at 4-6 hpi. Successfully developed invasion hypha (IH) play crucially roles in the time of later infected phase after 24 hpi. In our study, we performed that CgENA1, CgENA2, CgENA3, and CgENA4 were significantly genes, which participate in development and differentiation of those infected structures. Therefore, transcript level of ENA genes showed two waves at 4-6 hpi and 24-36 hpi at the significantly times of infection.

2 Question: Regarding to CgEna in potassium and sodium uptake (Figure 2): I think potassium and sodium mycelia concentration corresponding to complemented Cg strains should be included to improve the results about role of Ena in ions uptake.

Response: Thank you for this concern! According to last reports, the Ena ATPase were involved in efflux of sodium and potassium (1). To confirmed this function, we tested the contents of sodium and potassium in the mycelial of WT and mutants. In this study, our purpose is evaluation whether the ENA genes are required for potassium and sodium uptake? We determined that K+ and Na+ concentration in the mycelia using a flame spectrophotometer as previously described (2, 3). These results indicate that CgEna1 and CgEna4 are involved in regulating sodium uptake in C. gloeosporioides. Therefore, it is not necessary to measure the K+ and Na+ concentration of complemented stains.

References:

(1) Haro, R.; Garciadeblas, B.; Rodríguez-Navarro, A. A novel P-type ATPase from yeast involved in sodium transport. FEBS. Lett. 1991, 291, 189-191.

(2) Pan, Y.T.; Li, L.W.; Yang, J.Y.; Li, B.; Zhang, Y.Z.; Wang, P.; Huang, L. Involvement of Protein Kinase CgSat4 in Potassium Uptake, Cation Tolerance, and Full Virulence in Colletotrichum gloeosporioides. Front. Plant Sci. 2022, 13, 873-898.

(3) Zhang, Y.Z.; Li, B.; Pan, Y.T.; Fang, Y.L.; Li, D.W.; Huang, L. Protein phosphatase CgPpz1 regulates potassium uptake, stress responses, and plant infection in Colletotrichum gloeosporioides. Phytopathology 2022, 112, 820-829.

3 Question: In figure 4: It seems to me that the lesions on C. lanceolata and L. chinensis × tulipifera, when infected by ΔCgena4 strain are more severe than lesions of complemented strains ΔCgena4/CgEna4 although the lesion size does not differ significantly when compared with WT strain. Please check it.

Response: Thank you for your comments. I changed this picture.

4 Question:Pg. 5, Line 178-179. “These data indicated that the deletion of CgEna3 delayed the conidial germination” According to the Fig. 5A, the deletion of CgEna1 gene delayed the conidial germination.

Response: Thank you! I have revised.

5 Question:Line 349-351. "In this study, the deletion of CgEna4 significantly inhibited the mycelium growth under combined conditions of 0.7 M sodium and pH 7 or pH 10, indicating that non-acidic pH may alter the ion resistance mediated by CgEna4 in C.gloeosporioides". In this sentence the authors could point out in a few words that the lack of ΔCgena4 function would affect the maintenance of redox homeostasis of transport of sodium in C. gloeosporioides as described in Results (Fig 2).

Response: Thank you for your suggestions. We have added words into discussion at lines 376-384 in new manuscript to explain this phenomenon in the manuscripts.

6 Question:“Abstract” Line 13, “Ena ATPases” The abbreviations should be spelling out the first time is cited.

Response:Thank your suggestion! I have explained “Ena ATPases” at the first time cited.

7 Question:“Material and Methods” Line 238, What mean “CM plates supplemented with 0.7 M NaCl and 0.7 M NaCl, respectively”?

Response:Thanks! We examined the full text carefully and revised.

8 Question: “Fungal mycelia were harvested and dried in a freeze dryer” Please describe the procedure to harvest the fungal mycelia

Response:Thanks! I have specifically introduced this experimental process at part of “Material and Methods” in details. All strains, including WT, mutants and their corresponding complemented strains cultured in liquid CM medium for 2 days. Then we collected the mycelium pellets through microcloth to remove the filtrate. Fungal mycelia were harvested and dried in a freeze dryer.

Reviewer 2 Report

In the submitted manuscript by Deng et al., entitled “Distinct roles of Ena ATP family proteins in sodium uptake, invasive growth, and full virulence in Colletotrichum gloeosporioides” the authors explored the role of Ena ATPase proteins in Na+ and K+ ion transport and their role in the virulence of Colletotrichum gloeosporioides.

Overall, the interpretations seem reasonable. The manuscript writing is understandable and easy to read, though there are a few instances where certain statements are ambiguous or not entirely correct. The findings from the study will help in understanding the role of Ena proteins in the pathogenesis of Colletotrichum gloeosporioides through manipulating ion transporters.

As I detail below, there are several issues that should be addressed.

1.     The authors mentioned sodium uptake in the title however the paper describes both sodium and potassium uptake. Please explain or change the title.

2.     Fig. 1: Authors claim that CgEna1/2/3/4 shows membrane localization, but they haven’t used any membrane specific marker to confirm the localization.

3.     It would be informative to provide complete information in figure legends wherever applicable. What does error bars represent, SD or SEM; statistical significance; how many replicates-biological or technical, etc.

4.     Fig. 4: Have you performed the growth curve of the deletion strains compared to WT? The difference in growth rates could affect the lesion length and symptom development in host.

Author Response

Reviewer # 2

1 Question:  The authors mentioned sodium uptake in the title however the paper describes both sodium and potassium uptake. Please explain or change the title.

Response:Thanks! It was reported that Ena ATPases were involved in efflux of sodium and potassium (1). This inspired us to compare to content of sodium and potassium in the WT and mutants. But our results showed that deletion of CgENA1, CgENA2, CgENA3, and CgENA4 had no significant effect on the content of mycelial potassium (Fig.2A). These results suggest that CgEna1, CgEna2, CgEna3, and CgEna4 are not involved in regulating potassium uptake in C. gloeosporioides. Therefore, we think that Ena proteins only are involved in efflux of sodium.

References:

  • Haro, R.; Garciadeblas, B.; Rodríguez-Navarro, A. A novel P-type ATPase from yeast involved in sodium transport. Lett. 1991, 291, 189-191.

2 Question:Fig. 1: Authors claim that CgEna1/2/3/4 shows membrane localization, but they haven’t used any membrane specific marker to confirm the localization.

Response:Thanks! To determine the cellular localization of ENA genes. We firstly construct a strain with a GFP-tag at the C-terminus of CgEna1/2/3/4-GFP into the CgEna1/2/3/4 mutants. Results obtained from microscopy revealed membrane localization of CgEna1/4. To exclude interference of cell wall, we obtained their protoplasts to observe their cell localization in constructed CgEna1/2/3/4-GFP strains. These results also confirmed CgEna1/2/3/4 expressed in cell membrane.

3 Question:Thanks! It would be informative to provide complete information in figure legends wherever applicable. What does error bars represent, SD or SEM; statistical significance; how many replicates-biological or technical, etc.

Response:Thanks! We have now provided complete information in all figure legends.

4 Question: Fig. 4: Have you performed the growth curve of the deletion strains compared to WT? The difference in growth rates could affect the lesion length and symptom development in host.

Response:Thanks! We evaluated the growth of ΔCgena1/2/3/4 mutants on complete medium (CM) plates. The mutants exhibited no obvious difference in vegetative hyphal growth (Figure S4A). The colony diameters of the ΔCgena1/2/3/4 mutants were also no obvious difference compared with the wild-type (SMCG1#C) and complemented strains (Figure S4B). Therefore, we think ENA genes are not involved in the regulation of growth in C. gloeosporioides. In addition, our results showed that infected structures, containing appressorium and invasive hyphal, were delayed in these mutants. Collectively, we suggestion that ENA genes were involved in development of infected structures to regulate pathogenicity rather than vegetative growth.

Round 2

Reviewer 2 Report

I have no further comments.